# Comparison of Mean Corneal Power of Annular Rings and Zones Using Swept-Source Optical Coherence Tomography

**DOI:** 10.3390/diagnostics12030754

**Published:** 2022-03-19

**Authors:** Jing Dong, Jinhan Yao, Shuimiao Chang, Piotr Kanclerz, Ramin Khoramnia, Xiaogang Wang

**Affiliations:** 1Department of Ophthalmology, The First Hospital of Shanxi Medical University, Taiyuan 030001, China; dongjing790610@163.com; 2Department of Cataract, Shanxi Eye Hospital Affiliated to Shanxi Medical University, Taiyuan 030002, China; yjh961106@163.com (J.Y.); changsm991101@163.com (S.C.); 3Hygeia Clinic, 80-286 Gdańsk, Poland; p.kanclerz@gumed.edu.pl; 4Helsinki Retina Research Group, University of Helsinki, 00014 Helsinki, Finland; 5The David J. Apple International Laboratory for Ocular Pathology, Department of Ophthalmology, University of Heidelberg, 69120 Heidelberg, Germany; ramin.khoramnia@med.uni-heidelberg.de

**Keywords:** anterior axial curvature, total cornea power, posterior axial curvature, zones, rings

## Abstract

This study aims to investigate differences in the mean corneal power of annular zones (corneal power measured over the inner annular zone of difference diameters) and rings (corneal power measured over a ring of different diameters) centered on the corneal apex using the swept-source optical coherence tomography technique. The mean anterior axial curvature (AAC), posterior axial curvature (PAC), and total corneal power (TCP) centered on the corneal apex with the annular rings (0–2 mm, 2–4 mm, 4–6 mm, and 6–8 mm) and zones were assessed using the ANTERION device. The paired-sample *t*-test was used for data comparison. For the 0–2 mm comparison, the AAC, PAC, and TCP values of rings and zones were interchangeable. For the 2–4 mm comparison, the AAC of the rings was lower than that of the zones (*p* = 0.004), and the TCP values of the rings were higher than that of the zones (*p* < 0.001). For the 4–6 mm comparison, the AAC of the rings was lower than that of the zones (*p* < 0.001), and the PAC and TCP values of the rings were higher than that of the zones (both *p* < 0.001). For the 6–8 mm comparison, the AAC of the rings was lower than that of the zones (*p* < 0.001), and the PAC and TCP values of the rings were higher than that of the zones (both *p* < 0.001). Comparisons between AAC and TCP in each sub-region showed significant differences both in the rings (*p* < 0.001) and the zones (*p* < 0.008). Differences in the AAC, PAC, and TCP measured at different diameters (2–4 mm, 4–6 mm, and 6–8 mm) of the rings and zones, centered on the corneal apex, should be noticed in clinical practice. As the diameter increases, the difference between the rings and the zones in terms of AAC, PAC, and TCP increase as well. Clinicians should also pay attention to differences between AAC and TCP for the rings and the zones within the same annular region.

## 1. Introduction

Phacoemulsification cataract surgery has evolved from a procedure aiming to replace an opaque lens to a surgery allowing to achieve excellent postoperative refractive outcomes. Hence, there is a growing need for greater accuracy in the calculation of intraocular lens (IOL) power. The most relevant parameters required for IOL power calculations include the ocular axial length and keratometric values, namely the anterior axial curvature (AAC), posterior axial curvature (PAC), and total corneal power (TCP). Of these, the accuracy of corneal curvature data has been improving with advances in measurement technology [1]. The continual improvement in measurement devices has also expanded the measurement range, increased the measurement accuracy, and enabled the acquisition of personalized data for different diameter rings (corneal power measured over rings of different diameters) or zones (corneal power measured over the inner annular zones with different diameters) to provide an overview of the homogeneity of the power distribution and refractive power of the entire cornea [2]. Moreover, the corneal power with different diameters of rings and zones provides cataract surgeons with important information in calculating IOL power using different formula and planning an IOL implantation and even for corneal refractive surgery planning.

Currently, both optical coherence tomography (OCT) and the Scheimpflug imaging system can evaluate the anterior and posterior corneal surface within the central 8 mm diameter of the cornea, thus providing valuable information for IOL power calculations after refractive corneal surgery [3,4,5]. Due to the slow scanning speed and poor repeatability, the time-domain OCT was hardly used alone to measure corneal power. In some platforms, e.g., the Visante OMNI (Carl Zeiss Meditec, Jena, AG), time-domain OCT had to be combined with Placido-ring topography to obtain the corneal power [6]. With OCT development, Fourier-domain OCT is able to provide better accuracy and repeatability for corneal power measurement without the combination of Placido-ring information on the anterior corneal surface [7]. However, a technical problem in these devices is the small imaging field, i.e., the scanning range of corneal keratometric data (e.g., 6 mm diameter in the Optovue SD-OCT (Freemont, CA, USA)) and fan distortion.

Except for the larger scanning range, swept-source OCT (SS-OCT) does not require optical correction or complex adjustments of the geometric parameters to prevent optical distortions [8]. Moreover, SS-OCT anterior segment tomography can provide a high-resolution cross-sectional image, greatly reduce the scanning time, and enhance the scanning depth and tissue penetration [4,9]. As corneal tomography devices abandoned the assumption of a fixed geometric relationship of the anterior corneal surface and posterior corneal surface, they have become a much more robust method for measuring corneal power, even in surgically modified eyes and pathological eyes [7]. As a high-resolution SS-OCT imaging device, the ANTERION Cornea App can measure the AAC, PAC, and TCP of the central 8 mm diameter of the rings or zones centered on the corneal vertex (Figure 1). Moreover, the corneal data from the ANTERION SS-OCT system can also be demonstrated as mean values within 2 mm-diameter spacing, including the central 0–2 mm, the 2–4 mm annular region, the 4–6 mm annular region, and the 6–8 mm annular region. Some studies have shown that the ANTERION has good reproducibility for ocular anterior segment biometry, which serves as a foundation for this study [10].

The AAC is calculated based on a refractive index of 1.3375 and according to the laws of Gaussian optics without considering the actual refractive effect of the posterior corneal surface. The AAC is routinely used for IOL power calculations and toric IOL planning. However, it does not take into account the calculation errors for toric IOLs, which may be caused by the actual curvature of the posterior corneal surface [11]. This shortcoming is of increasing interest to clinicians.

The posterior axial curvature, using the actual refractive index of the cornea (*n* = 1.376) and aqueous humor (*n* = 1.336), is also measured according to the laws of Gaussian optics. Studies have found that the average PAC measured in the normal cornea using the Scheimpflug imaging system is approximately −6.29 D [12]. Furthermore, PAC can affect IOL power and toric IOL calculations [11]. Therefore, the current online formulas (e.g., Barrett Toric) and IOL manufacturers have all incorporated actual measured PAC data into their calculations [13].

The total corneal power is calculated using ray tracing technology that determines how parallel light beams are refracted based on the true refractive indices of the cornea and aqueous humor, the slope of the cornea, and the exact point of refraction. TCP uses the actual refractive indices of the different refractive media, while also accounting for the impact of the actual PAC. Therefore, TCP is not only included in the conventional formulae for IOL power calculation but is also used for IOL power calculations after previous corneal refractive surgery for myopia or hyperopia [14,15].

With various corneal power measurement data available, surgeons need to be aware of the potential differences between them. Indeed, it is difficult for ophthalmologists to decide which parameter is the most suitable for IOL calculations. In this prospective observational study, we compared the AAC, PAC, and TCP values for different annular zones and rings centered on the corneal apex using the ANTERION app in 90 routine cataractous eyes planned for surgery.

## 2. Materials and Methods

### 2.1. Subjects

This study was performed at the Shanxi Eye Hospital (Taiyuan, Shanxi, China). The research protocol was approved by the institutional review board of Shanxi Eye Hospital. The study was carried out according to the tenets of the Declaration of Helsinki. Written informed consent was obtained from each subject after explaining the nature of this study. This observational study has been registered online (at the International Standard Randomized Controlled Trials at http://www.controlled-trials.com (accessed on 8 November 2021)) with the registration number: ISRCTN13860301.

Consecutive prospective patients scheduled for cataract surgery were enrolled between November 2020 and March 2021. Inclusion criteria were as follows: routine cataract patients in our clinic, no systemic disease, no pathological alteration of the anterior segment (such as keratoconus, corneal opacity, or dry eye), no retinal diseases impairing visual function, and no previous anterior or posterior segment surgery. Exclusion criteria were as follows: ocular surface abnormality or pathology, history of eye trauma, contact lens wear, previous corneal refractive surgery, regular use of any eye drops, or unstable fixation on the target during imaging.

### 2.2. Sample Size

The sample size for the paired-sample *t*-test was calculated using MedCalc software (Version 20.014, MedCalc Software Ltd, Ostend, Belgium.). The type I error (Alpha, Significance) was set as 0.05, and the type II error (Beta, 1-Power) was set as 0.20. Based on our previous corneal power comparison result, the input value of the mean anterior corneal power difference was 0.06, and the standard deviation of anterior corneal power differences was 0.18. After calculation, the minimum required number was 73 eyes [2].

### 2.3. Data Acquisition

Corneal keratometry was measured using the ANTERION SS-OCT system (Heidelberg Engineering, Germany, software version 1.2.3.0) with the “Cornea” app for each eye in automatic release mode. The standard cornea scan mode consisting of maximum default 65 radial scanning lines centered on the cornea apex with a scan length of 9 mm was used for this study. Two consecutive measurements were captured for each eye. Measurements with good acquisition quality (checking parameters including eye tracking, motion, fixation, tear film and lid, camera image segmentation, refraction correction, required data points) were used in the final analysis. All measurements were performed in a semi-dark room, and no medication was used to dilate the pupils. The subjects were asked to place their chin on the chin rest and press their forehead against the forehead strap. The eye was then aligned to the visual axis by using a central fixation target. The subjects were instructed to perform a complete blink before each measurement. A single trained operator performed all examinations.

### 2.4. AAC, PAC, TCP Definition

The AAC, PAC, and TCP values were calculated based on the central 8 mm-diameter cross-sectional corneal image centered on the corneal vertex (Figure 2). The calculation method was introduced as follows [10]:

#### 2.4.1. AAC

Without considering the corneal refractive effect and the radii of the posterior corneal surface. The standard keratometric index of 1.3375 was used to make the conversion of anterior corneal radii to the keratometry data.

#### 2.4.2. PAC

Considering the refractive indices of the cornea (1.376) and the aqueous humor (1.336), the posterior corneal axial curvature data were calculated.

#### 2.4.3. TCP

Considering the slope of the cornea, the exact point of refraction and the true refractive indices of the cornea (1.376) and the aqueous humor (1.336), the TCP was calculated using the ray-tracing method.

### 2.5. Statistics

Statistical analyses were performed with commercial software (SPSS, ver. 13.0; SPSS Inc., Chicago, IL, USA). The mean values for the AAC, PAC, and TCP were computed. The Kolmogorov–Smirnov test was used to assess data normality. A paired two-tailed *t*-test was performed to check whether there was a significant difference between corneal power values centered on the corneal apex in different ring and zone diameters. The same statistical method was used to detect the mean difference between the AAC and TCP of different zones and rings centered on the cornea apex. All tests had a significance level of 5%.

## 3. Results

In total, 90 patients (90 eyes) who visited our clinic for cataracts were enrolled in the study. The study included 42 females and 48 males, with a mean age of 57 ± 18 years, a mean axial length of 24.04 ± 1.97 mm, and a mean central corneal thickness of 530 ± 38 μm. As shown in Table 1, regarding the AAC, data for the rings and zones were completely consistent within the 2 mm range, whereas for the 2–4 mm, 4–6 mm, and 6–8 mm comparisons, data for the zones were all higher than the corresponding data for the rings (all *p* < 0.005). Moreover, the standard deviations for AAC, PAC, and TCP were all relatively large in the current study.

The difference between the two gradually increased with increasing diameters (Figure 3a). In terms of the PAC, data for the rings and zones were also completely consistent within the 2 mm range, and no significant difference was found for the data comparison of 2–4 mm annular region, whereas for the 4–6 mm and 6–8 mm comparisons, data for the zones were all lower than the corresponding data for the rings (all *p* < 0.001). The difference between the two gradually increased with increasing diameters (Figure 3b). In terms of the TCP, data for the rings and zones were completely consistent within the 2 mm range, whereas for the 2–4 mm, 4–6 mm, and 6–8 mm comparisons, data for the zones were all lower than the corresponding data for the rings (all *p* < 0.001), and the difference between the two gradually increased with increasing diameters (Figure 3c).

Additionally, we found statistically significant differences between the AAC and TCP for the rings in different annular regions. The mean difference ranged from −1.73 D to 0.90 D (Table 2, Figure 4). Similarly, significant differences were found in the zones of different annular regions, and the mean difference ranged from −0.71 D to 0.90 D (Table 2, Figure 4). Comparing the different sub-regions, the smallest difference in the AAC and TCP between the rings and zones was found in the 4–6 mm annular range.

## 4. Discussion

Corneal refractive power is one of the key factors affecting IOL power calculations. Advances in ocular biometric technology have led to substantial improvements in the range and accuracy of corneal refractive power measurements. In this study, we employed the SS-OCT (a biometric technology for measuring the anterior segment) to measure and compare the differences in the AAC, PAC, and TCP between the rings and zones for different annular regions, and to compare the differences between the AAC and TCP of different diameter zones and rings, which can serve as data support for the calculation and analysis of IOL power. Our findings indicated that within the 0–2 mm annular region, the AAC, PAC, and TCP were completely consistent between the zones and rings. However, within the 2–4 mm, 4–6 mm, and 6–8 mm annular regions, the AAC of the rings was smaller than that of the zones. Conversely, the PAC and TCP of the rings were larger than those of the zones. Comparisons between the AAC and TCP for the rings in different annular regions showed significant differences between the two, which increased with increasing measurement diameter. Significant differences were also found between the AAC and TCP for the zones of different annular regions.

Our findings indicated that within the 0–2 mm annular region, the AAC, PAC, and TCP were completely consistent between the rings and zones. These results implied that the data acquired using the two methods were completely interchangeable within the 2 mm region. Moreover, it also showed that there were no significant morphological changes between the anterior and posterior corneal surfaces within this range. As the range of comparison expanded towards the periphery, we detected a significant increase in the differences between the rings and zones for all data compared. In terms of the AAC, the average difference between the two ranged from 0.01 D to 0.36 D. Based on the SRK/T formula for IOL power calculation (IOL power = IOL A constant − 2.5 ∗ axial length − 0.9 ∗ average keratometry), the difference in keratometry data between the two may lead to a difference in IOL power ranging from 0.009 D to 0.324 D. For an IOL interval of 0.5D, the difference between rings and zones was not clinically significant for the calculated IOL power. However, as the IOL interval gradually became finer from 0.5D to 0.25D, this difference became increasingly clinically significant [16,17]. Regarding the TCP, the difference between the rings and zones ranged between 0.06 D and 0.65 D. Based on the same inference as above, this difference in keratometry may lead to a difference in the IOL power ranging from 0.054 D to 0.585 D. Although the difference in IOL power for the 6–8 mm diameter was greater than 0.5 D, the keratometry range for IOL power tended to be within 4 mm; hence, the difference between the AAC and TCP was not clinically significant.

Furthermore, the difference between AAC and TCP reflected the optical calculation principles and the refractive indices used. For both the rings and zones, AAC showed a decreasing trend with increasing diameter, suggesting that the corneal curvature becomes flatter as the range of measurement expands toward the periphery. This is consistent with the results by Scott et al. [18]. However, the weight of the actual PAC was also captured. In contrast, taking into consideration the true refractive indices of the posterior corneal surface and the different refractive media, TCP showed an opposing trend of change to AAC. This further reflected the impact of PAC on the total corneal refractive state. When the data analysis was limited to the rings of different annular regions, a significant difference was found between AAC and TCP, and the range of difference was relatively large (−1.73 D to 0.90 D). This demonstrated that PAC has a substantial impact on TCP data [19]. Additionally, when the data analysis was limited to the zones of different annular regions, we found a large range of differences between AAC and TCP (−0.71 D to 0.90 D). This further suggests that PAC significantly influenced the TCP and that this influence is of particular importance in the calculations for IOLs [20,21].

### Strengths and Limitations

Compared to the time-domain and Fourier-domain OCT imaging technology, SS-OCT is able to provide a larger area of actual corneal power and pachymetry map information. Apart from the SS-OCT imaging technology used in this study, the Scheimpflug imaging method can also provide keratometric values, such as AAC, true net power, and TCP within different diameter rings and zones [22,23,24]. In a recent study, Scheimpflug and OCT imaging were both able to detect tomographic patterns of subclinical corneal edema in patients with Fuchs endothelial corneal dystrophy [25]. Devices based on SS-OCT and Scheimpflug technologies are corneal tomographers. They are capable of assessing the corneal power homogeneity of the entire cornea and provide necessary information for IOL power calculation in eyes with previous corneal refractive surgery. The ring and zone comparison findings in this study with the ANTERION device may also indicate a change in mean corneal power in Scheimpflug devices. The current comparison was only performed in relatively healthy cornea. The relatively large range of ages in the current study may have contributed to the large variation in comparison data. Therefore, further studies in keratoconus screening, IOL power calculation, and patients with previous corneal refractive surgery as well as normal subjects with bigger sample sizes are required. As an extension to the current study, performing axial scan analysis of the cross-sectional images along the depth of each subject can be helpful for an in-depth analysis in the thickness variations and as a multilayered comparative study among subjects.

## 5. Conclusions

In conclusion, our data demonstrated a significant difference in AAC, PAC, and TCP, measured at different diameters (2–4 mm, 4–6 mm, and 6–8 mm) of the rings and zones centered on the corneal apex. With an increase in diameter, the difference in AAC, PAC, and TCP between the rings and zones also increased. Moreover, there were significant differences between AAC and TCP in both the zones and rings for the same annular region.

## Figures and Tables

**Figure 1 diagnostics-12-00754-f001:**
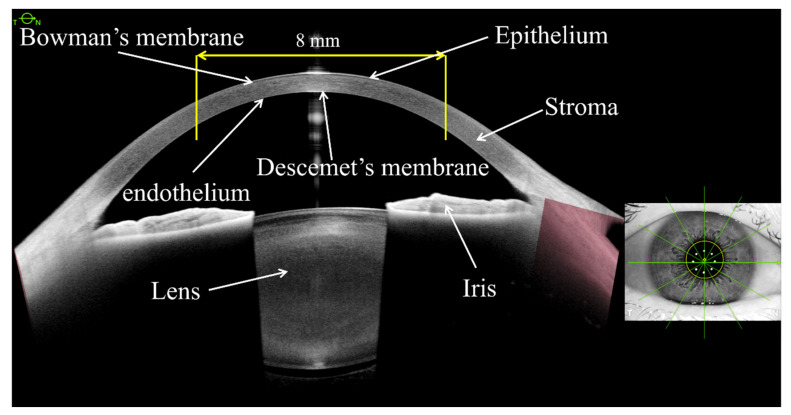
The cross-sectional image and corresponding ocular tissue information of anterior segment and corresponding scanning directions on the eye image using ANTERION device. The central 8 mm-diameter cornea area centered on corneal vertex based on 65 radial scanning lines was used for corneal curvature analysis.

**Figure 2 diagnostics-12-00754-f002:**
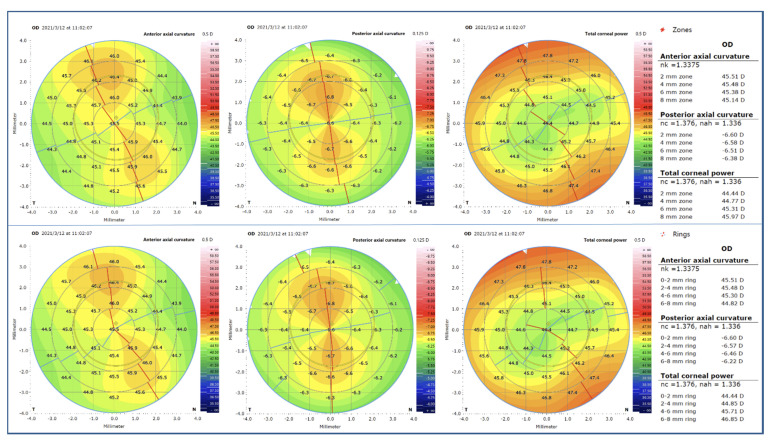
The anterior axial curvature, posterior corneal curvature, and total corneal power changes with different ring and zone diameters centered on the corneal vertex for a right eye. The first row was segmented by different diameter zones, and the second row was segmented by different diameter rings. The red and blue lines represent the meridian of steep and flat corneal curvature, respectively.

**Figure 3 diagnostics-12-00754-f003:**
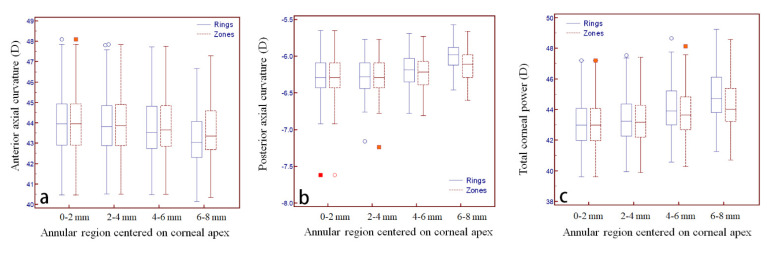
Box and whisker plot of the anterior axial curvature (**a**), posterior axial curvature (**b**), and total corneal power (**c**) changes with different rings and zones diameters centered on the corneal apex.

**Figure 4 diagnostics-12-00754-f004:**
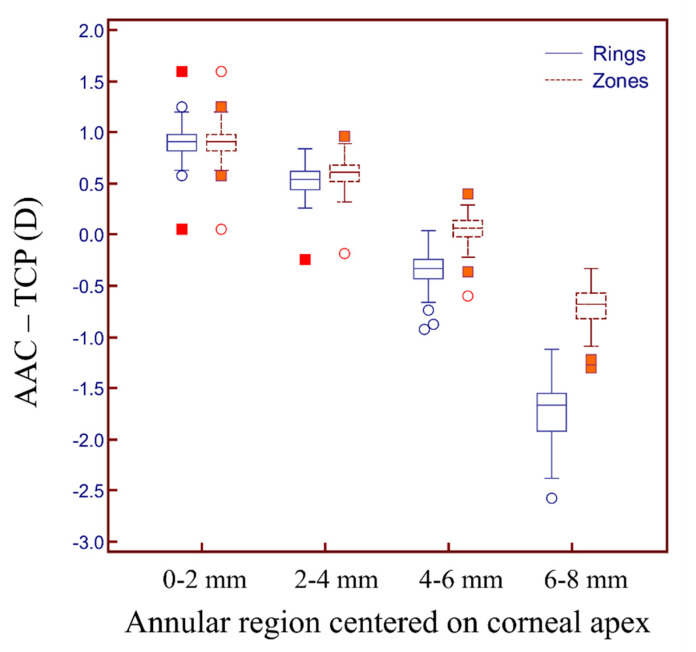
Box and whisker plot of AAC and TCP difference for rings and zones in different annular regions.

**Table 1 diagnostics-12-00754-t001:** Mean anterior axial curvature, posterior axial curvature, and total corneal power of different zones and rings centered on the apex.

	Rings (M ± SD)	Zones (M ± SD)	Rings-Zones (M ± SE)	*p* *
Anterior axial curvature (D)				
0–2 mm diameter	44.02 ± 1.58	44.02 ± 1.58	0.00 ± 0.00	N/A
2–4 mm diameter	43.95 ± 1.52	43.97 ± 1.53	−0.01 ± 0.004	**0.004**
4–6 mm diameter	43.71 ± 1.44	43.82 ± 1.47	−0.11 ± 0.015	**<0.001**
6–8 mm diameter	43.18 ± 1.34	43.54 ± 1.40	−0.36 ± 0.025	**<0.001**
Posterior axial curvature (D)				
0–2 mm diameter	−6.27 ± 0.28	−6.27 ± 0.28	0.00 ± 0.00	N/A
2–4 mm diameter	−6.26 ± 0.25	−6.26 ± 0.25	0.001 ± 0.002	0.450
4–6 mm diameter	−6.19 ± 0.23	−6.22 ± 0.23	0.03 ± 0.004	**<0.001**
6–8 mm diameter	−5.99 ± 0.19	−6.12 ± 0.21	0.13 ± 0.01	**<0.001**
Total corneal power (D)				
0–2 mm diameter	43.11 ± 1.59	43.11 ± 1.59	0.00 ± 0.00	N/A
2–4 mm diameter	43.43 ± 1.56	43.37 ± 1.56	0.06 ± 0.01	**<0.001**
4–6 mm diameter	44.07 ± 1.57	43.77 ± 1.55	0.30 ± 0.02	**<0.001**
6–8 mm diameter	44.91 ± 1.59	44.26 ± 1.55	0.65 ± 0.03	**<0.001**

Note: D = diopter; M = mean; N/A = not applicable; SD = standard deviation; SE = standard error. * Paired two-tailed *t*-test. Statistically significant values (at the 5% level) are in bold.

**Table 2 diagnostics-12-00754-t002:** Mean difference between anterior axial curvature and total corneal power of different zones and rings centered on apex.

AAC-TCP (D)	Rings (M ± SD)	*p* *	Zones (M ± SD)	*p* *
0–2 mm diameter	0.90 ± 0.17	<0.001	0.90 ± 0.17	**<0.001**
2–4 mm diameter	0.53 ± 0.15	<0.001	0.60 ± 0.15	**<0.001**
4–6 mm diameter	−0.36 ± 0.18	<0.001	0.04 ± 0.15	**0.007**
6–8 mm diameter	−1.73 ± 0.28	<0.001	−0.71 ± 0.19	**<0.001**

Note: D = diopter; M = mean; SD = standard deviation. * Paired two-tailed *t*-test. Statistically significant values (at the 5% level) are in bold.

## Data Availability

The data presented in this study are available in Appendix A.

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
