# Peer review of "Comparison of Mean Corneal Power of Annular Rings and Zones Using Swept-Source Optical Coherence Tomography"

_diagnostics, 2022, doi:10.3390/diagnostics12030754_

Round 1
Reviewer 1 Report
The manuscript is focused on the comparison of mean corneal power of annular rings and zones using swept-source optical coherence tomography (SS-OCT). In the abstract, "A This study aims to investigate differences in the mean corneal", please correct the "A This". It will help a lot if the researcher could provide the images by SS-OCT. In the figure, for example the figure 1, the variation looks large. Could the researcher explain the reason? The technical writing can be improved. The literature searching should be improved. "Major Revision" is recommended before being considered.
Author Response
Response to the reviewers’ comments
We are very pleased that the revised version of our paper is being considered for publication in the Diagnostics. Please find below our point-by-point responses to the comments of the reviewers:
The manuscript is focused on the comparison of mean corneal power of annular rings and zones using swept-source optical coherence tomography (SS-OCT).
In the abstract, "A This study aims to investigate differences in the mean corneal", please correct the "A This". It will help a lot if the researcher could provide the images by SS-OCT.
Answer: We made the correction. Line 14.
In the figure, for example the figure 1, the variation looks large. Could the researcher explain the reason?
Answer: We speculated that this big variable may attribute to the relatively big age range from 23 to 90 years old in this study. We also added corresponding information in the limitation part. Line 293-296.
The technical writing can be improved. The literature searching should be improved. "Major Revision" is recommended before being considered.
Answer: We updated the technical writing by using the language editing service from Editage. Moreover, we added some additional references. Line 55-75, 149-161.
We thank the reviewer for his constructive comments. The manuscript ultimately looks better in its present form and we hope it will be found suitable for publication.
Sincerely,
Xiaogang Wang

Reviewer 2 Report
Overall, the study seems to be interesting and the application of using OCT for corneal anterior and posterior axial curvature measurements and corneal power measurements is a widely growing and well-accredited use of OCT in clinical and research applications. But the authors have not given enough detailed background explanations in the introduction highlighting the superiority of using OCT for cornea assessment. Furthermore, another functional use of different OCT systems for cornea analysis reported by other research groups is not discussed adequately mentioned. Following are other key information that are required from the reviewer’s point of view which needs to be addressed in order for the manuscript to be considered for publication. Minor English corrections are required. For example, the abstract starts as “A this study…” Even though the manuscript is sufficiently clear for readers to understand the author's intensions, few articles and grammar checks can improve the manuscript readability.
Comments:
- The patient age group used as subjects in this study has a very wide range of ± 18 years. This seems too big, It is understandable that finding subjects within a short range of age-group is difficult. In which case, a short note explaining this difficulty should be addressed.
- How AAC, PAC, and TCP measurements are made is not well explained, It is understandable that the authors use a commercial system for measurement, but this does not levi the authors to simply ignore how the measurements are executed. Either the authors can give a detailed explanation of how an ANTERION system calculates the AAC, PAC, and TCP. The other option is to include explanations of how other research groups have previously measured, this can be included in introduction section.
- It would benefit the readers if the authors include a reference curvature map (figurative) for both AAC & PAC.
- Why haven’t the authors included an en face, cross-section, and 3D volume representative images of a patient with necessary layer naming?
- The authors should also consider including the section area of where and how the AAC and PAC regions are calculated from OCT images.
- The standard deviations of Figure 2 and figure 3 are too big. The authors have addressed this, but the explanations and possible factors leading to this should be further explained discussion and also briefly note in the results section.
Author Response
Response to the reviewers’ comments
We are very pleased that the revised version of our paper is being considered for publication in the Diagnostics. Please find below our point-by-point responses to the comments of the reviewers:
Overall, the study seems to be interesting and the application of using OCT for corneal anterior and posterior axial curvature measurements and corneal power measurements is a widely growing and well-accredited use of OCT in clinical and research applications. But the authors have not given enough detailed background explanations in the introduction highlighting the superiority of using OCT for cornea assessment. Furthermore, another functional use of different OCT systems for cornea analysis reported by other research groups is not discussed adequately mentioned. Following are other key information that are required from the reviewer’s point of view which needs to be addressed in order for the manuscript to be considered for publication. Minor English corrections are required. For example, the abstract starts as “A this study…” Even though the manuscript is sufficiently clear for readers to understand the author's intensions, few articles and grammar checks can improve the manuscript readability.
Comments:
- The patient age group used as subjects in this study has a very wide range of ± 18 years. This seems too big, It is understandable that finding subjects within a short range of age-group is difficult. In which case, a short note explaining this difficulty should be addressed.
Answer: We emphasize the potential relationship between big range age and relatively large data variation and added corresponding information in the limitation part. Line 293-295.
- How AAC, PAC, and TCP measurements are made is not well explained, It is understandable that the authors use a commercial system for measurement, but this does not levi the authors to simply ignore how the measurements are executed. Either the authors can give a detailed explanation of how an ANTERION system calculates the AAC, PAC, and TCP. The other option is to include explanations of how other research groups have previously measured, this can be included in introduction section.
Answer: in order to enrich the AAC, PAC and TCP information, we added more information in the method part. Line 148-161.
- It would benefit the readers if the authors include a reference curvature map (figurative) for both AAC & PAC.
Answer: we added figure 2 to demonstrate this information. Line 162-167.
- Why haven’t the authors included an en face, cross-section, and 3D volume representative images of a patient with necessary layer naming?
Answer: we added figure 1 to demonstrate this information. Line 83-86. However, our current device version could not provide the 3D volume information.
- The authors should also consider including the section area of where and how the AAC and PAC regions are calculated from OCT images.
Answer: we added this section and information in the method part. Line 148-161.
- The standard deviations of Figure 2 and figure 3 are too big. The authors have addressed this, but the explanations and possible factors leading to this should be further explained discussion and also briefly note in the results section.
Answer: we added the corresponding information in the results and discussion part. Line 181-184; 293-298.
We thank the reviewer for his constructive comments. The manuscript ultimately looks better in its present form and we hope it will be found suitable for publication.
Sincerely,
Xiaogang Wang

Round 2
Reviewer 2 Report
The authors have addressed the queries raised in the previous review round.
Provided that the authors address the following comments, the manuscript can be considered for publication.
Comments:
- In section "2.3 Data Acquisition, line 139", the authors have mentioned they used 65 radial scanning lines, with a scan length of 9 mm. Why did they use such a less number of scanning lines? Does this mean the Gap between two b-scans is 130 micrometers? If so, then this is more than 10 times the resolution of the OCT system. Why didn't the authors use more sampling points (higher B-scan number)? Higher sampling will aid in more detailed structures in samples with higher intensities. This needs to be addressed in the methods section.
- Merge Figures 3,4, and 5 into one figure with 3a, 3b, and 3c. Results observed and reported are too few.
- The result section is very brief, the authors should consider improving the results section by trying to incorporate more analysis from the obtained datasets, for example, they can perform an axial scan analysis of the cross-section image (centered 2D scan) along the depth of each subject to analyze the thickness variation. This will improve the overall observed result analyzed and the quality of the manuscript.
Author Response
Response to the reviewers’ comments
We are very pleased that the revised version of our paper is being considered for publication in the Diagnostics. Please find below our point-by-point responses to the comments of the reviewers.
We thank the reviewer for his constructive comments. The manuscript ultimately looks better in its present form and we hope it will be found suitable for publication.
Sincerely,
Xiaogang Wang
- In section "2.3 Data Acquisition, line 139", the authors have mentioned they used 65 radial scanning lines, with a scan length of 9 mm. Why did they use such a less number of scanning lines? Does this mean the Gap between two b-scans is 130 micrometers? If so, then this is more than 10 times the resolution of the OCT system. Why didn't the authors use more sampling points (higher B-scan number)? Higher sampling will aid in more detailed structures in samples with higher intensities. This needs to be addressed in the methods section.
Answer: As the reviewer mentioned, we also believe that more sampling points will provide more details about the corneal structure. However, for current ANTERION device version of the corneal analysis, the maximum default scanning line intensity was 65 radial scanning lines, which the user is unable to make change of it (see image below and we consulted the manufacturer about this information). Line 138-140.
- Merge Figures 3,4, and 5 into one figure with 3a, 3b, and 3c. Results observed and reported are too few.
Answer: the merged figure 3 was updated in the revised version. Line 207-210.
Figure 3. Box and whisker plot of the anterior axial curvature (a), posterior axial curvature (b), and total corneal power (c) changes with different rings and zones diameters centered on the corneal apex.
- The result section is very brief, the authors should consider improving the results section by trying to incorporate more analysis from the obtained datasets, for example, they can perform an axial scan analysis of the cross-section image (centered 2D scan) along the depth of each subject to analyze the thickness variation. This will improve the overall observed result analyzed and the quality of the manuscript.
Answer: This is a very interesting aspect to enrich the results section. Based on our current device version, it cannot provide the thickness values for rings and zones of different diameters, and we need to develop specific segmentation algorithm to extract these data from the 65 2D scanning images for each eye, which we may consider in our further study.
We can provide the central corneal thickness (CCT) values into the results part. Line 178-180. No significant correlation was found between CCT and all three mean corneal power data of central 2 mm zones and rings (all P > 0.05). Therefore, we did not add this information into the result part.

Round 3
Reviewer 2 Report
The authors have addressed the comments raised in previous review rounds.
Though the authors have not further analyzed the results by using a segmentation or intensity and thickness assessment algorithm, the current manuscript can be used as a base for further research by the authors or other research groups for further in-depth analysis by extending the current study.
From a reviewer's point of view, it is strongly recommended that the authors include a sentence in the conclusion and in discussion sections stating,
"As an extension to the current study, performing axial scan analysis of the cross-section images along the depth of each subject can be helpful for an in-depth analysis in the thickness variations and as a multilayered comparative study among subjects."
Given the authors address the above comment, the manuscript in its current form can be considered for publication.
Author Response
1. The authors have addressed the comments raised in previous review rounds.
Though the authors have not further analyzed the results by using a segmentation or intensity and thickness assessment algorithm, the current manuscript can be used as a base for further research by the authors or other research groups for further in-depth analysis by extending the current study.
From a reviewer's point of view, it is strongly recommended that the authors include a sentence in the conclusion and in discussion sections stating,
"As an extension to the current study, performing axial scan analysis of the cross-section images along the depth of each subject can be helpful for an in-depth analysis in the thickness variations and as a multilayered comparative study among subjects."
Given the authors address the above comment, the manuscript in its current form can be considered for publication.
Answer: We added the corresponding information about the necessity of axial scan analysis of corneal thickness variation investigation in the discussion part, and we will do further study about it. Line 307-310.